# Distinct homotypic B-cell receptor interactions shape the outcome of chronic lymphocytic leukaemia

Claudia Minici[1,2,*], Maria Gounari[3,*,†], Rudolf Übelhart[4], Lydia Scarfò[2,3,5], Marcus Dühren-von Minden[4], Dunja Schneider[6], Alpaslan Tasdogan[4], Alabbas Alkhatib[6], Andreas Agathangelidis[3], Stavroula Ntoufa[7], Nicholas Chiorazzi[8], Hassan Jumaa[4], Kostas Stamatopoulos[7,9], Paolo Ghia[2,3,5] & Massimo Degano[1]

Cell-autonomous B-cell receptor (BcR)-mediated signalling is a hallmark feature of the neoplastic B lymphocytes in chronic lymphocytic leukaemia (CLL). Here we elucidate the structural basis of autonomous activation of CLL B cells, showing that BcR immunoglobulins initiate intracellular signalling through homotypic interactions between epitopes that are specific for each subgroup of patients with homogeneous clinicobiological profiles. The molecular details of the BcR–BcR interactions apparently dictate the clinical course of disease, with stronger affinities and longer half-lives in indolent cases, and weaker, short-lived contacts mediating the aggressive ones. The diversity of homotypic BcR contacts leading to cell-autonomous signalling reconciles the existence of a shared pathogenic mechanism with the biological and clinical heterogeneity of CLL and offers opportunities for innovative treatment strategies.

[1] Biocrystallography Unit, Division of Immunology, Transplantation and Infectious Diseases, IRCCS San Raffaele Scientific Institute, via Olgettina 58, 20132 Milan, Italy. [2] Università Vita-Salute San Raffaele, 20132 Milan, Italy. [3] B-cell Neoplasia Unit, Division of Experimental Oncology, IRCCS San Raffaele Scientific Institute, 20132 Milan, Italy. [4] Universitätsklinik Ulm, 89081 Ulm, Germany. [5] Strategic Research Program on CLL, IRCCS San Raffaele Scientific Institute, 20132 Milan, Italy. [6] Centre for Biological Signaling Studies, Faculty of Biology, Albert-Ludwigs University of Freiburg, 79104 Freiburg, Germany. [7] Institute of Applied Biosciences, Center for Research and Technology, 57001 Thessaloniki, Greece. [8] The Feinstein Institute for Medical Research, Manhasset, New York 11030, USA. [9] Department of Immunology, Genetics and Pathology, Uppsala University, 75105 Uppsala, Sweden. * These authors contributed equally to this work. † Present address: Institute of Applied Biosciences, Center for Research and Technology, 57001 Thessaloniki, Greece. Correspondence and requests for materials should be addressed to P.G. (email: ghia.paolo@hsr.it) or to M.D. (degano.massimo@hsr.it).

Chronic lymphocytic leukaemia (CLL) is the most common adult leukaemia in the West, characterized by a monoclonal expansion of mature, antigen-experienced $CD5^+$ B cells[1,2]. CLL is a highly heterogeneous disease both in terms of biological features[3], with leukaemic B cells ranging from functionally anergic to highly proliferating, and clinical courses varying from indolent to highly aggressive.

Signalling initiated by antigen binding to B-cell receptor (BcR) immunoglobulins (IGs) is of paramount importance throughout the natural history of the disease[4]. Indeed, BcR signalling pathways are constitutively active in all CLL cases[5,6], and inhibitors of the downstream effectors Bruton's tyrosine kinase (Ibrutinib) or phosphoinositide 3-kinase δ (Idelalisib) show clinical efficacy[7,8]. This evidence complements earlier observations supporting antigen drive in CLL ontogeny, including the distinction of CLL into cases with somatically hypermutated BcR IG ('mutated' CLL) who have a significantly better outcome compared to those with unmutated receptors ('unmutated' CLL)[9,10]. CLL cells also display a remarkably skewed BcR IG gene repertoire, culminating in the existence of highly homologous, stereotyped receptors in more than 30% of cases[11,12], indicating selection by restricted antigenic elements.

Interestingly, CLL cases expressing stereotyped BcR IGs can be categorized into several subsets, each of them displaying highly homogeneous biological features[3] and clinical presentation and outcome[11,13–15]. For instance, CLL subset no. 4 shows a particularly indolent clinical course that is linked to a characteristic anergic functional phenotype of the malignant cells[16]. Subset no. 4 clones express γ-switched BcR IGs encoded by the *IGHV4-34/IGKV2-30* gene pair[11,17] with somatic hypermutation (SHM) patterns similar to edited autoantibodies[18]. On the opposite extreme, CLL stereotyped subset no. 2 is noted for a dismal prognosis, largely independent of p53 dysfunction[13,14,19,20]. The corresponding CLL cells express immunoglobulin-M (IgM) BcR encoded by the *IGHV3-21/IGLV3-21* gene pair[21], all displaying distinctive SHMs[18]. Thus, stereotyped CLL cases recapitulate the overall features of the disease, in the context of homogeneous biological and clinical behaviour within individual subsets.

Despite the proven role of antigenic stimulation in CLL ontogeny, the nature of the molecular antigens involved in leukaemic cell selection and stimulation remains controversial. Recently, a cell-autonomous model of CLL cell activation was demonstrated whereby BcR IGs from both stereotyped and non-stereotyped CLL cases could promote $Ca^{2+}$ influx and nuclear factor-κB target gene transcription without the addition of exogenous antigen. This signalling was proposed to occur through the recognition of a single, common BcR epitope conserved in all cases[22].

The existence of a single, unifying activation mechanism for all CLL B cells needs to be reconciled with the known heterogeneity of the disease and the differences in cellular responsiveness to external stimuli. To this end, we investigated the relevant molecular interactions underlying cell-autonomous signalling in CLL cases with opposite biological features and clinical course. We demonstrate that BcR IGs derived from both indolent and aggressive CLL cases interact homotypically via their combining sites, binding to distinct internal epitopes in each subset of patients. These unexpected BcR–BcR interactions are sufficient to initiate intracellular $Ca^{2+}$ influx, and mediate stronger affinities and long half-lives for the receptors derived from anergic B cells from indolent clinical cases, while weaker and short-lived for the aggressive ones. The amino-acid residues involved in the homotypic BcR interactions are acquired as a consequence of class-switch recombination or SHMs that shape each epitope towards complementarity with the combining site. Our results offer a molecular and structural basis for the clinicobiological heterogeneity of CLL, unveiling the diversity of homotypic BcR contacts in the context of cell-autonomous signalling by CLL-derived BcR IGs and linking the quality of the BcR signal to the distinct clinical outcomes. These findings open the way to the development of specific treatments based on BcR antagonism.

## Results

**CLL subset no. 4 Fabs can interact homotypically**. We determined the crystal structures of the fragments involved in antigen binding (Fab) derived from the BcR IG of the subset no. 4 cases CLL183 and CLL240 (ref. 17), representing functionally anergic and clinically indolent CLL (Fig. 1a and Table 1)[16]. The VH and VK complementary determining regions (CDRs) of these clonotypic BcR IGs concur to form an undulating surface reminiscent of anti-protein antibodies[23]. In both CLL183 and CLL240 Fab crystals (Fig. 1a, Supplementary Fig. 1 and Supplementary Fig. 2), the molecules form a lattice characterized by specific intermolecular interactions involving the VH CDR3 loop of one receptor and an articulate surface comprising residues from the heavy chain framework region 1 (VH FR1) and the CH1 domain of a second Fab (Fig. 1b). The Fab surface areas buried on complex formation (558 and 597 $Å^2$ for CLL183 and CLL240, respectively) are within the range for antibodies in complex with their specific, high-affinity protein antigens (Supplementary Table 1). Distinctive features of the interaction are a salt bridge between the CDR3H residue $Arg^{107H}$ and $Glu^{16H}$ from VH FR1, the insertion of the side chain of $Tyr^{108H}$ in a hydrophobic pocket, hydrogen bonds and van der Waals' contacts between residues $Tyr^{109H}$, $Tyr^{110H}$ and $Tyr^{111H}$ and the surface of the CH1 domain (Fig. 1b). The light chain contributes to a lesser extent to the interaction surface, establishing only two direct hydrogen bonds through residues $Tyr^{54L}$ and $Ser^{61L}$ from the VK CDR2 loop. Thus, the intermolecular interactions observed in subset no. 4 BcR crystals are highly indicative of a specific homotypic recognition primarily mediated by VH CDR3 and directed towards a conformational epitope spanning the VH and CH1 domains.

**Self-association of CLL subset no. 4 Fabs in solution**. Next, we assessed whether the interaction observed between the subset no. 4 BcR IGs is representative of the behaviour of the molecules in solution. Using analytical ultracentrifugation (AUC), we observed that both Fabs from the CLL183 and CLL240 receptors displayed concentration-dependent association in solution (Fig. 2a). Sedimentation velocity (SV) experiments demonstrated the presence of a single species with sedimentation coefficient 3.7 S at low protein concentration, and the concentration-dependent appearance of a second species (5.0 S) consistent with the formation of a dimeric assembly. Then, we performed a sedimentation equilibrium (SE) analysis that was optimally fitted with a two-state self-association model (Fig. 2b, Supplementary Fig. 3 and Supplementary Table 2), with $K_D$ values in the low μM range. The dissociation rate constants $k_{off}$ obtained from the SV AUC data are indicative of long-lived, stable binding interactions (Table 2).

Next, we engineered and expressed in recombinant form two mutants of the CLL183 Fab designed to disrupt the intermolecular interactions observed in the crystal structures. No self-association could be detected for either the CLL183 mutant Fabs $Glu^{16H}Ala$ (mutated at a key amino acid of the epitope, indicated FR) and $Arg^{107H}Ala/Tyr^{108H}Ala$ (bearing a double mutation in the paratope region, indicated CDR) (Fig. 2c,d), thus confirming that the contacts observed in the experimental crystal structures recapitulate the actual interactions occurring in solution.

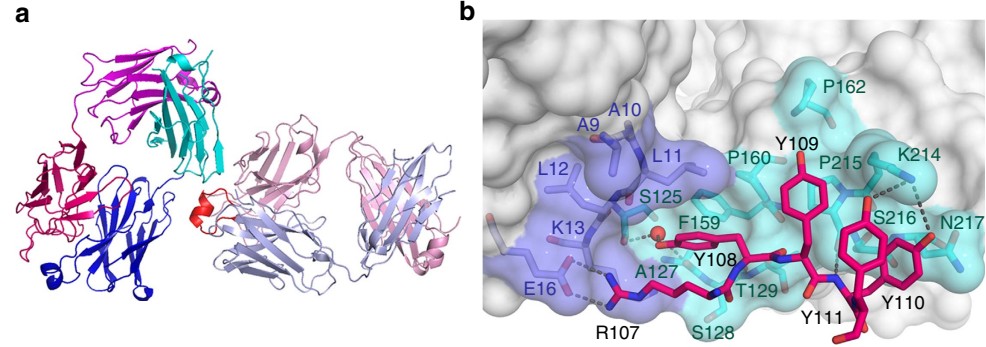

**Figure 1 | CLL subset no. 4 Fabs interact homotypically with antibody–antigen features.** (**a**) The Fab molecules in both the CLL183 and CLL240 crystals interact through VH CDR3-dominated contacts. The heavy and light chains of the Fab acting as 'receptor' are coloured light blue and pink, with the VH CDR3 loop highlighted in red. The chains in the 'antigen' molecule are coloured blue and cyan (VH and CH1 domains), and violet and magenta (VK and CK), respectively. (**b**) Molecular surface of the epitope, with the amino acids from VH CDR3 involved in the binding interactions shown as sticks coloured as in the previous panel. Parts of the structure have been omitted for clarity.

| Table 1 \| Data collection and refinement statistics. | | | |
|---|---|---|---|
| | **CLL240** | **CLL183** | **P11475** |
| *Data collection* | | | |
| Unit-cell parameters (Å) | 86.19, 92.33, 135.13 | 90.21, 90.21, 131.23 | 65.87, 65.87, 111.23 |
| Space group | $P2_12_12_1$ | $P4_32_12$ | $P4_1$ |
| Molecules (a.u.) | 2 | 1 | 1 |
| Solvent fraction | 0.555 | 0.552 | 0.500 |
| Resolution range (Å) | 46.20–2.10 (2.23–2.10) | 42.66–2.27 (2.34–2.27) | 65.87–2.29 (2.37–2.29) |
| Unique reflections, $I(hkl) > -3\sigma$ | 63,033 (9,838) | 25,817 (2,237) | 21,207 (1,930) |
| Completeness (%) | 99.7 (98.1) | 100 (100) | 99.3 (93.0) |
| Redundancy | 13.3 (12.3) | 8.6 (9.1) | 7.1 (7.4) |
| $<I/\sigma(I)>$ | 12.3 (1.3) | 10.7 (2.2) | 121.7 (2.9) |
| CC(1/2) (%) | 99.9 (66.8) | 99.8 (50.4) | 99.8 (41.2) |
| $R_{meas}$ | 0.14 (1.28) | 0.127 (1.21) | 0.141 (2.074) |
| $R_{pim}$ | 0.041 (0.657) | 0.042 (0.390) | 0.052 (0.753) |
| | | | |
| *Refinement* | | | |
| Resolution range (Å) | 46.16–2.10 | 42.66–2.27 | 65.87–2.29 |
| Reflections used, $F(hkl) > 0$ (free) | 63,020 (3,194) | 25,766 (1,319) | 21,028 (1,078) |
| $R_{crys}/R_{free}$ | 0.194/0.222 | 0.195/0.241 | 0.179/0.209 |
| No. of atoms | | | |
| Protein | 6,853 | 3,352 | 2,956 |
| Water | 647 | 90 | 70 |
| B-factors (Å$^2$) | | | |
| Protein | 47.8 | 60.0 | 62.0 |
| Water | 42.5 | 46.7 | 44.1 |
| R.m.s. deviations | | | |
| Bond lengths (Å) | 0.009 | 0.007 | 0.004 |
| Bond angles (°) | 1.060 | 1.036 | 0.750 |

**Subset no. 4 BcR self-recognition initiates signalling**. Since each BcR carries two epitopes and combining sites, the observed interactions may lead to receptor declustering and crosslinking on the CLL B-cell surface[24,25] to activate cell-autonomous intracellular signalling[22]. We thus expressed four different subset no. 4 CLL-derived BcRs in a *RAG2/λ5/SLP65* triple knockout murine pre-B-cell line (TKO cells)[26] engineered to include a tamoxifen-inducible ER$^{T2}$-SLP65 fusion protein[22]. On addition of 4-hydroxy tamoxifen and no exogenous antigen, TKO cells expressing the wild-type BcRs displayed an increase in Ca$^{2+}$ influx that is diagnostic of cell-autonomous signalling (Fig. 3). Conversely, cells expressing the epitope or the paratope BcR IG mutant did not demonstrate significant variations in Ca$^{2+}$ influx. Both the wild-type and mutant BcRs were responsive to crosslinking, thus demonstrating their retained signalling

function (Fig. 3). Mutation to alanine of the VH CDR3 residue Thr$^{102H}$, an amino acid exposed to solvent but not involved directly in homotypic BcR interaction, did not affect signalling (Supplementary Fig. 4). Hence, the CLL subset no. 4 IgG receptors are endowed with a combining site that allows specific, long-lived, high-affinity self-recognition through VH CDR3-mediated interactions with a conformational epitope contained within the BcR IG itself, and this binding leads to B-cell activation.

**Recurrent SHMs strengthen self-binding in subset no. 4 BcRs**. A notable feature of stereotyped subset no. 4 concerns the presence of 'subset-biased', recurrent amino-acid changes introduced by SHM at specific positions of the IG sequence. This

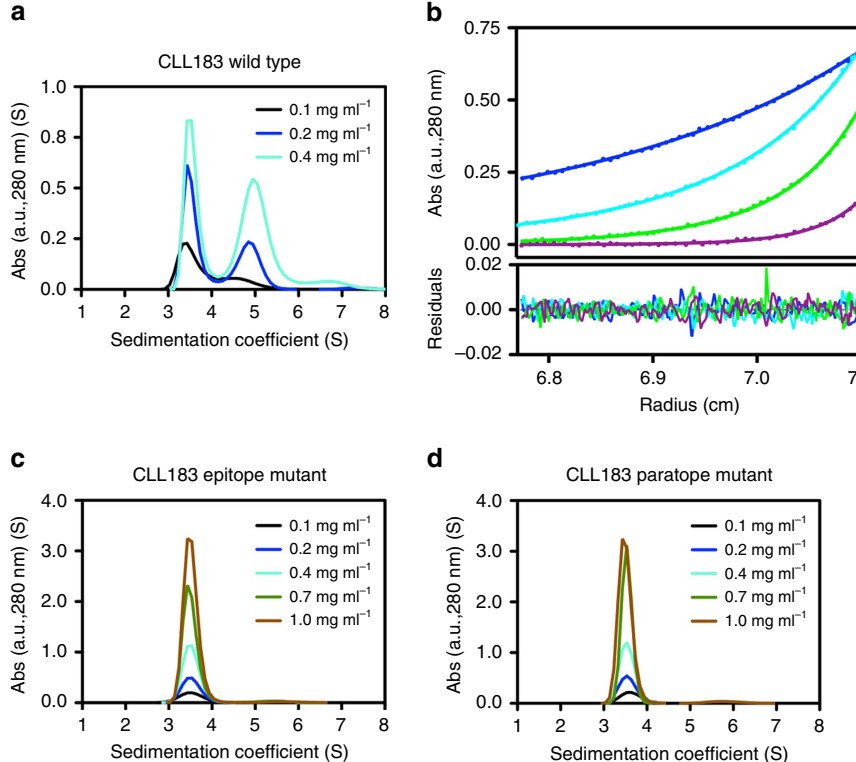

**Figure 2 | Characterization of subset no. 4 homotypic interactions in solution.** (**a**) Analytical ultracentrifugation sedimentation velocity analysis of the CLL183 Fab. Two primary species are present in solution with sedimentation coefficients of 3.7 and 5.0 S, respectively ($1 S = 10^{-13} s$). The appearance of the larger species is concentration dependent, consistent with the presence of a monomer–dimer equilibrium. (**b**) Fitting of SE-AUC data with a self-association model. The points represent the absorbance at a specific position in the cell, and the curve show the fitting of the chosen model. In this experiment, the protein concentration was $0.26 \, mg \, ml^{-1}$, and the cell was subjected to speeds of 8,000 (blue points and line), 12,000 (cyan), 16,000 (green) and 22,000 r.p.m. (violet). The residuals are small and show no systematic trends (see also Supplementary Fig. 3 and Supplementary Table 3). (**c**,**d**) For both the CLL183 Fab epitope (Glu[16H]Ala) and paratope (Arg[106H]Ala/Tyr[107H]Ala) mutants a single, monomeric species is present in solution even at protein concentrations of $1 \, mg \, ml^{-1}$ (corresponding to $20 \, \mu M$).

**Table 2 | Distinct structural properties of CLL BcR IGs homotypic interactions.**

| BcR IG (subset) | BSA ($Å^2$) | $N_{HB}$ | $N_{SB}$ | BE (kcal mol$^{-1}$) | Sc | $K_D^*$ ($\mu M$) | $k_{off}^{\dagger}$ (s$^{-1}$) |
|---|---|---|---|---|---|---|---|
| CLL183 (4) | 558 | 5 | 4 | −4.2 | 0.730 | 15.6 ± 0.8 | $(1.99 \pm 0.09) \times 10^{-5}$ |
| CLL240 (4) | 597 | 5 | 4 | −5.1 | 0.724 | 14.0 ± 0.4 | $(2.63 \pm 0.08) \times 10^{-4}$ |
| P11475 (2) | 557 | 1 | 4 | −0.2 | 0.680 | 430 ± 160 | $>10^{-2}$ |

BE, binding energy, computed from the experimental structures using the PISA server; BSA, buried surface area; $K_D$, equilibrium dissociation constant; $k_{off}$, dissociation kinetic rate constant; $N_{HB}$ and $N_{SB}$, number of direct hydrogen bonds and salt bridges established at the protein–protein interface; Sc, surface complementarity.
*Experimental values are shown as mean ± s.d.
†Experimental values are shown as mean ± s.d.

is exemplified by the characteristic substitution Tyr[31L]His introduced in the IGKV2-30 germline sequence[27]. This amino acid is part of a hydrogen bonding network that ultimately stabilizes the aromatic cage and Arg[106H], both crucial for the binding site architecture. The IGKV2-30 germline-encoded Tyr[31L] would induce a repositioning of the Arg[106H] side chain, and affect the VH CDR3 conformation (Supplementary Fig. 5a). Furthermore, the recurrent Gln[43L]His substitution[27] localized in the VK FR2 optimizes the VH–VK interactions, and is hence affecting the active site architecture. Similar effects can be seen by the Gly[31H]Glu mutation in the heavy chain, since the germline-encoded Gly is present in the CLL183 receptor (Supplementary Fig. 5b), while the mutated Glu is present in the CLL240 IG (Supplementary Fig. 5c). The acidic Glu reside enhances the interaction surface between paratope and epitope, and stabilizes the intramolecular interaction (Table 2). Taken together, SHMs

apparently synergize to stabilize the combining site structure and thus the self-interaction shown by subset no. 4 IGs.

**Class switch is needed for subset no. 4 autonomous signalling.** CLL subset no. 4 BcRs are exclusively IgG switched[11,28]. A close inspection of the epitope provides a structural rationale for this observation (Fig. 4a). Indeed, residue Lys[214H], whose side chain in the IgG CH1 domain forms hydrogen bonds that stabilize the self-interaction, is not conserved in IgMs or IgEs (Fig. 4b). We engineered a chimeric BcR where the constant domains of the CLL183 BcR IG were swapped with the murine Cγ1/Cκ pair. This receptor lacks residue Lys[214H], and indeed does not retain the autonomous signalling property of the wild-type BcR (Fig. 4c). Hence, subset no. 4 CLL BcR IGs acquire their self-recognition capacity on class-switch recombination to IgG.

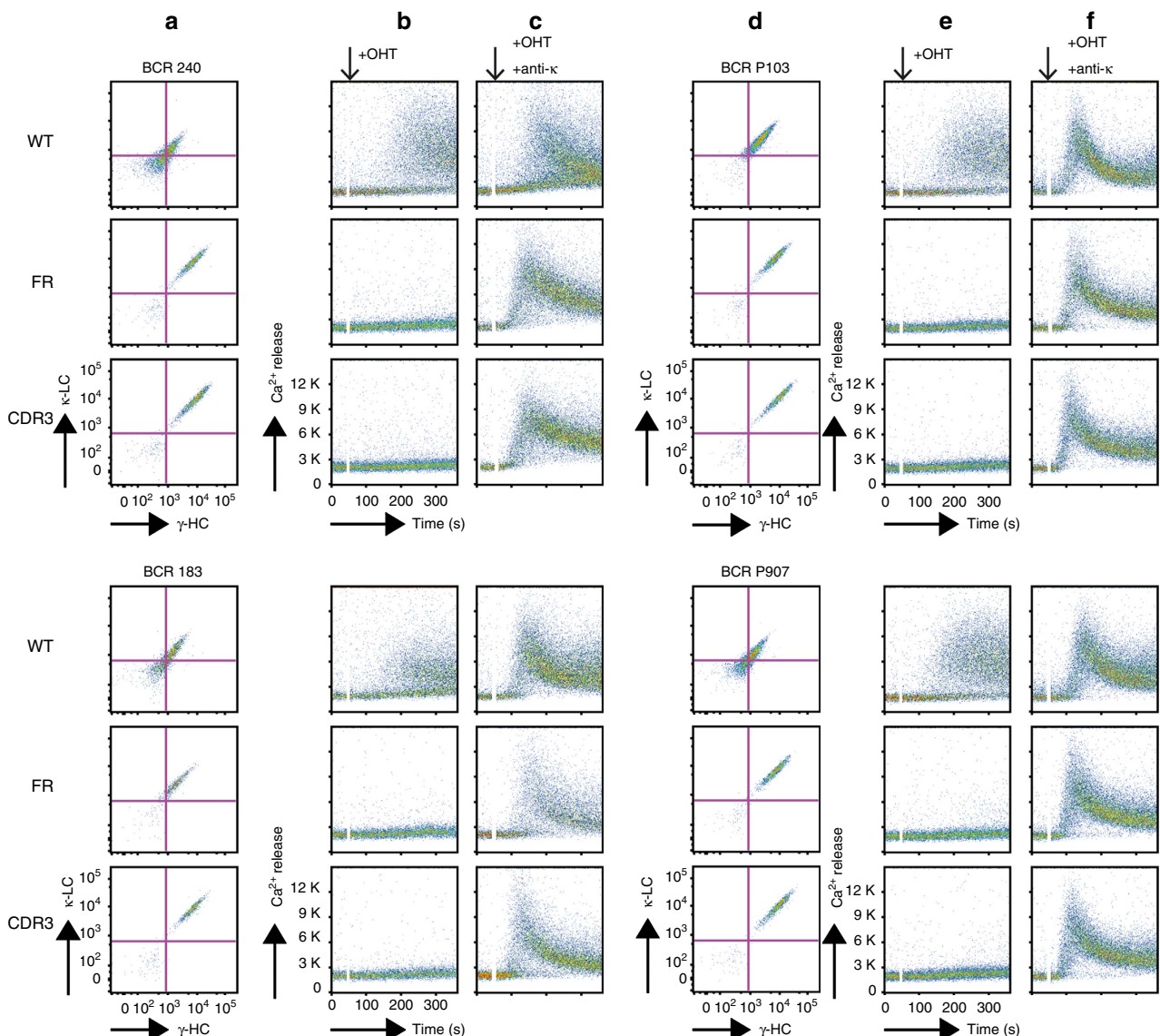

**Figure 3 | Subset no. 4 BcR homotypic binding leads to intracellular signalling in B cells. (a,d)** Four subset no. 4 CLL-derived clonotypic BcR IGs (CLL183, CLL240, P103, P907) were expressed in TKO cells reconstituted with a OHT-inducible variant of SLP-65 (ER$^{T2}$-SLP65), and IG surface expression was assessed. **(b,e)** Intracellular Ca$^{2+}$ influx on induction of SLP65 function with OHT. While the wild-type (WT) receptors lead to Ca$^{2+}$ influx, neither the Glu$^{16H}$Ala (FR) nor the Arg$^{106H}$Ala/Tyr$^{107H}$Ala (CDR3) mutants signalled on OHT addition. **(c,f)** All receptors expressed in the TKO cells were competent for signalling, and induced Ca$^{2+}$ influx when crosslinked with an anti-κ-chain antibody. Addition of the stimuli is indicated by black arrows.

**Light-chain-mediated contacts between CLL subset no. 2 BcRs.** Next, we determined the crystal structure of the Fab fragment derived from the IgM-expressing, clinically aggressive CLL subset no. 2 clone P11475 (Fig. 5a and Table 1). Similarly to subset no. 4, the intermolecular interactions in the crystal (Fig. 5a) are suggestive of antibody binding to antigen (Table 2 and Supplementary Table 1). Two symmetry-related molecules in the P11475 crystals interact through the VL CDR1 and VL CDR2 loops and a composite region spanning the FR1 and linker region between the VL and CL domains of the light chain (Fig. 5b). Charge–charge interactions are established between residues Arg$^{110L}$ in the VL–CL linker region and Asp$^{50L}$ in the VL CDR2 loop. Residue Lys$^{16L}$ in the FR1 region of the λ-chain protrudes between the VL CDR1 and VL CDR2 loop, establishing a salt bridge with Asp$^{52L}$, and an interaction with the helix dipole of VL CDR1. The characteristically short VH CDR3 allows a close approach between the two receptors, but establishes only one direct hydrogen bond with the epitope through residue Asn$^{101H}$.

The homotypic interaction for the subset no. 2 BcR IGs is apparently weaker compared to that in subset no. 4. Indeed, despite resulting in comparable buried surface areas and surface complementarity[29], it involves fewer polar intermolecular contacts (Table 2). Computational calculations of the binding energy also suggest a weak interaction, and indeed dimeric species of the P11475 Fab could be detected in SV AUC experiments only at a protein concentration of 20 μM (Fig. 5c).

Since the high absorbance of more concentrated P11475 solutions prevented the measurement of $K_D$, we resorted to dilution isothermal titration calorimetry to obtain a dissociation constant of 430 ± 160 μM (Fig. 5d). This value is close to the maximum concentration attainable for the P11475 Fab in solution, 300 μM, and thus still limiting the sensitivity of mutagenesis studies in solution. Hence, we used the TKO cell system to assess the physiological relevance of the homotypic interaction observed in the subset no. 2 BcR IGs crystals. Two distinct BcR IGs derived from CLL subset no. 2 cases promoted

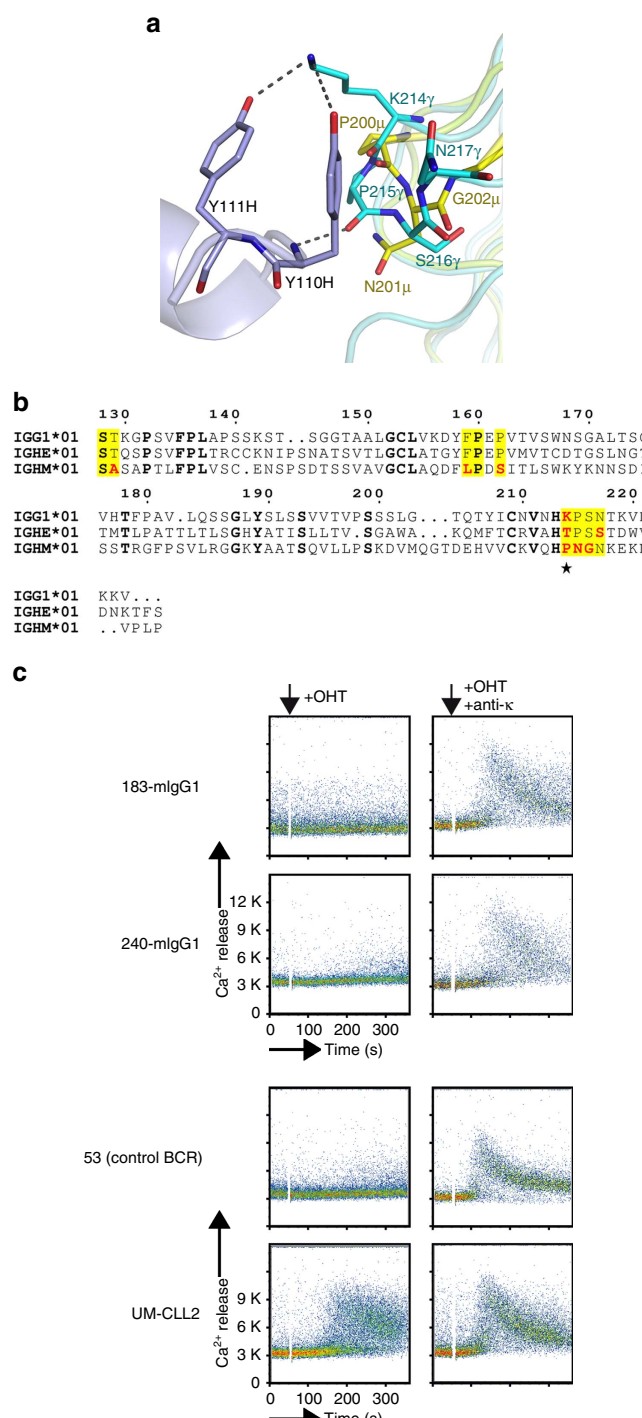

**Figure 4 | Class-switch recombination to IgG introduces the binding epitope in subset no. 4 CLL BcR IGs.** (**a**) Superposition of Cγ1 and Cμ1 domain structures. The different amino-acid sequence and conformation of the μ-chain constant domain (coloured yellow) compared to correspondent region in the γ-chain results in the loss of hydrogen bonds and polar contacts with subset no. 4 VH CDR3, thus hampering the formation of the homotypic interactions. The positively charged amino acid Lys[214H], involved in hydrogen bonding interactions with the BcR combining site, is not conserved in either IgE or IgM CH1 domains. Moreover, the conformation of the –PNG– loop in Cμ1 prevents the formation of a hydrogen bond to the main chain nitrogen of Tyr[110H], and of van der Waals contacts between Tyr[109H] and the domain surface. (**b**) Amino-acid sequence alignment for IgG, IgE and IgM constant domains. The residues that contribute to the conformational epitope recognized by subset no. 4 CLL BcR IGs are highlighted in yellow, and in red when endowed with different chemical properties. (**c**) Substitution of the wild-type Cγ1/Cκ domains with the mouse IgG1 leads to loss of autonomous signalling to the subset no. 4 BcRs derived from the CLL240 (CLL240-mIgG1) and CLL183 (CLL183-mIgG1) cases. The mouse Cγ1 mutant lacks the side chain of Lys[214H] substituted with a Pro, and is thus unable to establish hydrogen bonds with the VH CDR3 residues Tyr[110H] and Tyr[111H] (see Fig. 1). Both mutated receptors signal on crosslinking with an anti-light-chain antibody. Positive and negative controls are a CLL-derived BcR that signals autonomously (UM-CLL2) and an unrelated BcR (53), respectively. Expression was similar for all receptors, as judged from fluorescence-activated cell sorting analysis.

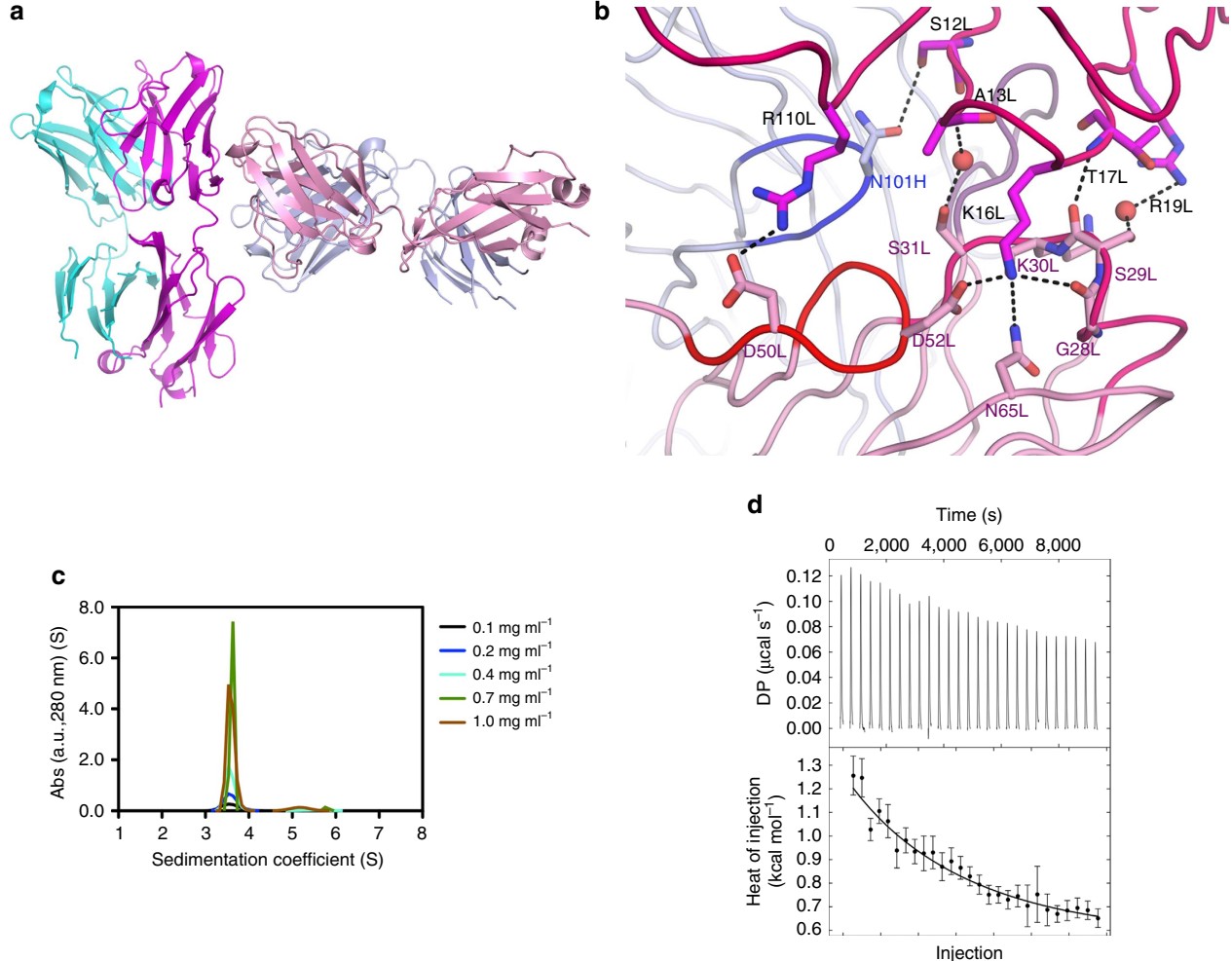

**Figure 5 | The CLL P11475 IgM Fab fragment interacts with a specific internal epitope in the light chain.** (**a**) The molecule acting as the receptor is coloured in light blue (heavy chain) and pink (light chain), respectively, with its long axis oriented horizontally. The heavy and light polypeptide chains in the antigen molecule are coloured cyan and violet, respectively. (**b**) The self-recognition by subset no. 2 BcR IGs is dominated by light-chain-mediated contacts. Two ordered solvent molecules, shown as red spheres, provide bridging interactions between epitope and paratope. Carbon atoms in the 'antigen' light chain are coloured violet, pink in the receptor light chain and light blue in the receptor heavy chain. (**c**) Analysis in solution of subset no. 2 BcR self-association. Dimeric species of the P11475 Fab (with expected sedimentation coefficient 5S) cannot be detected at protein concentrations of 20 μM in SV AUC experiments, consistent with a low-affinity interaction and fast dissociation rate. (**d**) Dilution ITC allow the detection of the weak homotypic complex. Top panel shows the raw calorimetric dilution titration profile for the subset no. 2 P11475 Fab fragment, the bottom panel the fit of the integrated heat of dissociation with $K_D = 430 \pm 160\,\mu M$ and $\Delta H^0 = -5.4 \pm 0.6\,\text{kcal mol}^{-1}$.

$Ca^{2+}$ influx when transduced in TKO cells (Fig. 6), and this property could be abolished only by the specific mutations (Asp[50L]Ala/Asp[52L]Ala in the paratope, Lys[16L]Ala, or Arg[110L]Gly in the epitope) engineered to disrupt the contacts observed in the crystal structure (Fig. 6). Instead, the mutation to alanine of residue Arg[19L] that neighbours the combining site but is positioned at 6 Å distance from possible interacting amino acids did not affect the autonomous signalling (Fig. 6). Thus, CLL subset no. 2 IgM BcR IGs can bind homotypically through their combining site to epitope residues that are distinct from the IgGs of CLL subset no. 4, and induce intracellular $Ca^{2+}$ influx.

**A unifying SHM leads to subset no. 2 BcR self-recognition.** The homotypic interactions between subset no. 2 BcRs are dominated by the *IGLV3-21*-encoded light chain, thus arguing for an ancillary role of the heavy chain in self-recognition and autonomous signalling. Indeed, the latter promotes the correct assembly of the functional BcR IG but provides limited contacts

with the epitope. Notably, *IGLV3-21* is the only light-chain gene germline product that bears both Asp residues in the VL CDR2 loop. Hence, the usage of the *IGLV3-21* gene in stereotyped subset no. 2 CLL cases is linked to its structural features that allow BcR–BcR interactions (Fig. 5b and Supplementary Fig. 6). The most recurrent SHM in the light chain of subset no. 2 CLL cases (Ser[93L]Gly, numbering according to the deposited P11475 structure) is positioned at the tip of the VL CDR3 loop with its Cα atom in contact with the opposite receptor, allowing a closer approach of the BcRs for homotypic interactions. Therefore, also the SHMs observed in the aggressive CLL subset no. 2 cases appear to modulate the self-interaction between BcR IGs.

The most striking finding stemming from the structural analysis of the subset no. 2 IgM is the crucial role in self-recognition of the unexpected Arg[110L] residue in the epitope (Fig. 5b), corresponding to the splice site between the *IGLJ3* and *IGLC* genes. This residue is present in all subset no. 2 BcR IGs sequenced ($N = 17$), arising from a single G->C nucleotide substitution, and its reversion to the genome-encoded Gly

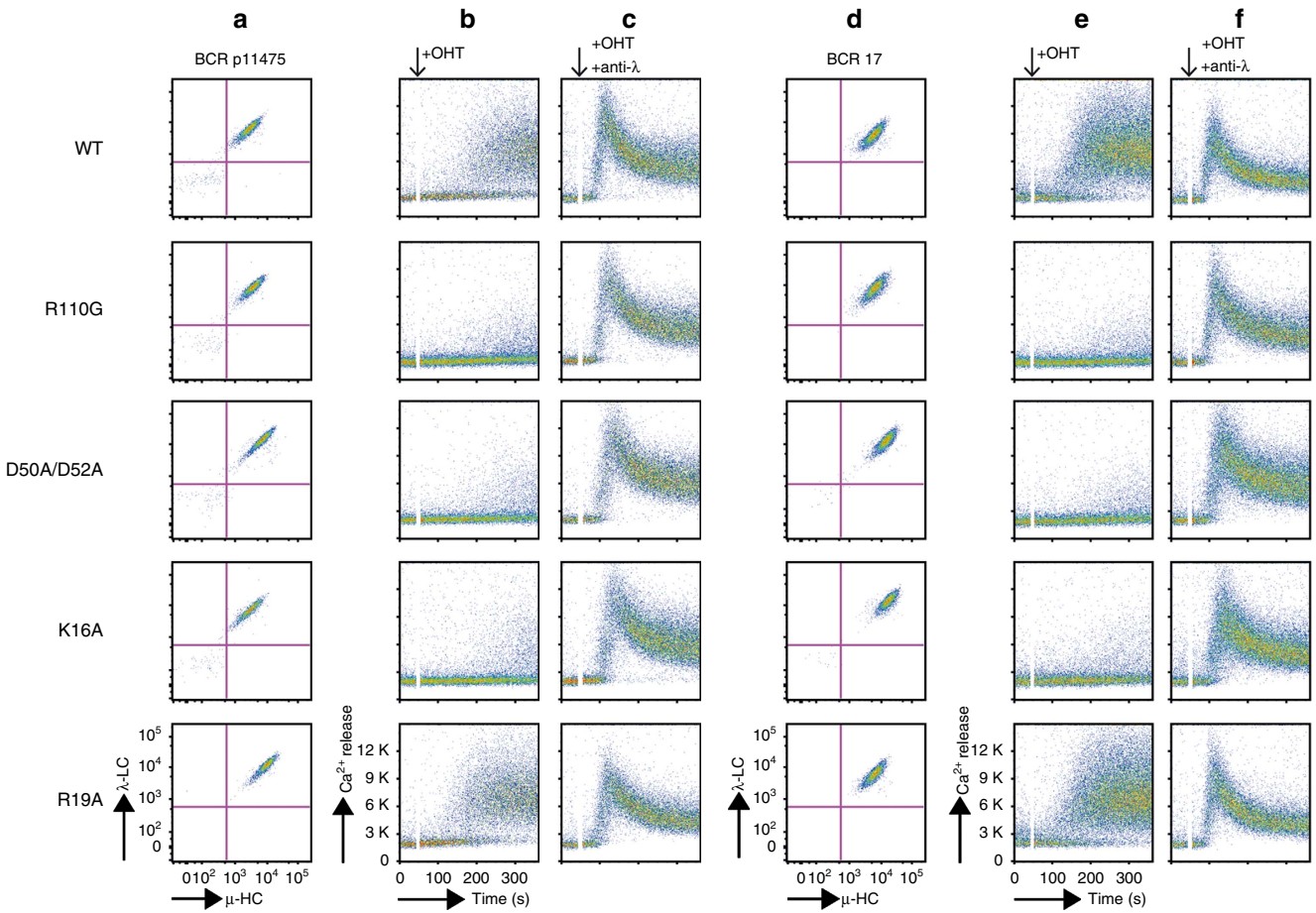

**Figure 6 | Autonomous signalling by CLL subset no. 2 IgM BcRs depends on a specific mutation from the germline sequence.** (**a,d**) Two subset no. 2 CLL-derived clonotypic BcR IGs (P11475, BCR17) were expressed in TKO cells reconstituted with a OHT-inducible variant of SLP-65 (ER$^{T2}$-SLP65). The IG surface expression is shown. (**b,e**) Intracellular Ca$^{2+}$ influx on induction of SLP65 function with OHT. The Arg$^{110L}$Gly (R100G), Asp$^{50L}$Ala/Asp$^{52L}$Ala (D50A/D52A) and Lys$^{16L}$Ala (K16A) mutants of both receptors did not signal on OHT addition. A control mutation (Arg$^{19L}$Ala) did not affect Ca$^{2+}$ influx. (**c,f**) When crosslinked with an anti-λ-chain antibody, both receptors induced Ca$^{2+}$ influx. In all panels showing Ca$^{2+}$ influx measurements, addition of OHT is indicated by black arrows.

abrogates cell-autonomous signalling (Fig. 6). Sequence analysis of DNA from non-leukaemic cells of subset no. 2 cases ($N = 7$) showed that the *IGLJ3* gene carries the germline G nucleotide, hence identifying the presence of the Arg residue in the BcR IG as a consequence of a true SHM rather than a genetic polymorphism. Thus, the ontogeny and evolution of CLL subset no. 2 cases seems inextricably linked to this specific, unifying SHM from the germline sequence in the stereotyped BcR IG.

## Discussion
Our results provide a description to high resolution of a homotypic association process in BcRs that resembles antibody–antigen recognition and leads to intracellular signalling in CLL cells. More interestingly, the internal epitopes recognized by CLL BcRs from different subgroups are distinct, and the avidity of the interaction explains the biological properties of the corresponding leukaemic cells, and may extend to the clinical features of each case. Thus, the self-recognition herein reported describes the molecular mechanism at the roots of cell-autonomous signalling in CLL, providing at the same time a receptor-specific molecular mechanism concurring to the typical heterogeneity of the disease.

The intermolecular interactions between the CLL BcRs can lead to receptor declustering at the cell surface, and thus the formation of a signalling-competent immune complex. The

location of the epitopes and the flexibility between the Fab and the Fc regions in BcR IG molecules[30] are compatible with both intracellular self-recognition (*cis*) and intercellular (*trans*) contacts[22] (a model for such interactions for IgGs is presented in Supplementary Fig. 7a,b). In the present study, we observed two distinct modes of self-recognition by BcR IGs from CLL cases belonging to paradigmatic subgroups with opposite clinical outcomes and functional features, both leading to the hallmark cell-autonomous signalling (Supplementary Fig. 8a,b). On these grounds, it is conceivable that each autoreactive CLL-derived BcR IG may bind to a distinct internal epitope, nevertheless leading to CLL cell activation. The present findings also demonstrate that the autonomous signalling in CLL is not a consequence of a generalized, intrinsic property of the preimmune BcR IG, but rather can also be acquired through distinct processes in IG affinity maturation, for example, class-switch recombination (as in subset no. 4) or the introduction of precise SHMs (as in subset no. 2 IgMs) that can be specific for each CLL case or subset thereof.

The self-recognition for subset no. 4 BcR IGs is tight and long-lived, and the subset-specific SHMs stabilize the receptor structure, enhancing the half-life of the signalling complex (Table 2) as expected in an interaction leading to B-cell anergy[31,32], which is indeed a feature of the clinically indolent subset no. 4 CLL clones[16,33,34]. Conversely, the aggressive clinical

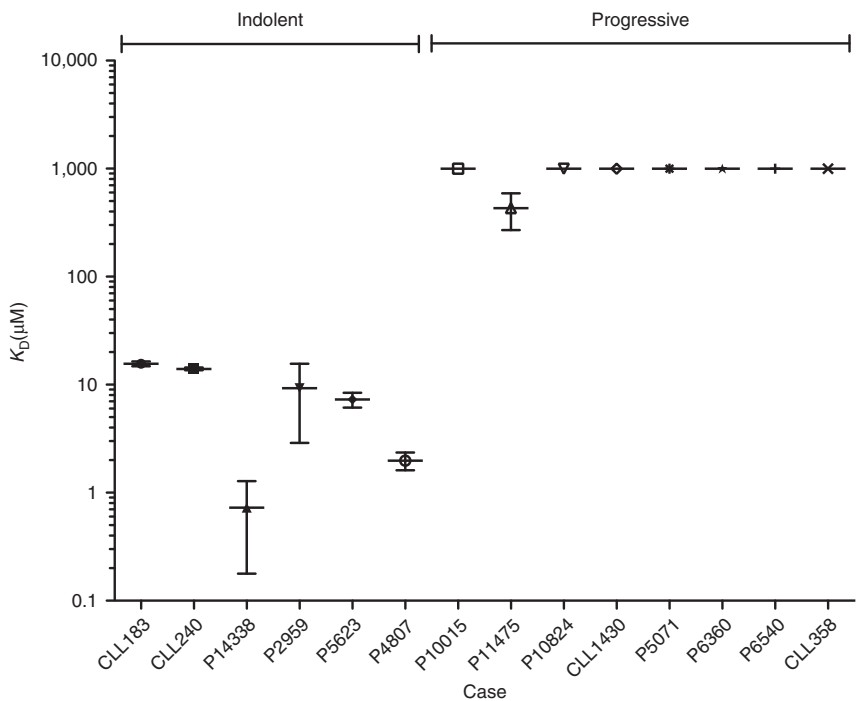

**Figure 7 | High-affinity homotypic interactions are characteristic of BcRs from indolent CLL cases.** The dissociation constants of Fab fragments of BcRs derived from CLL cases were measured using SV-AUC and dilution ITC. Fab fragments from five indolent cases displayed $K_D$ values in the 0.7–16 µM range. Among the eight aggressive cases, the $K_D$ value for P11475 was 430 ± 160 µM (see Fig. 5), while the remaining Fabs did not show dimeric species at protein concentration up to 20 µM in SV-AUC experiments (plotted at 1,000 µM for visualization purposes). Under the experimental conditions used, the appearance of a dimer peak in the $c(S)$ distribution derived from SV-AUC experiments required at least $k_{off} < 10^{-2} s^{-1}$ and $K_D < 500$ µM (based on simulations with SEDFIT and SEDPHAT). Data are means of three experiments, error bars are standard deviations.

course in subset no. 2 cases is paralleled by a low-affinity and fast-dissociating BcR IG self-recognition (Table 2), and a steady CLL cell response on receptor crosslinking that reflects a higher reactivity through the BcR[35]. It is thus conceivable that differences in affinity and half-life of the homotypic BcR complex allow serial triggering of the receptor that leads to a qualitatively different activation of intracellular signalling cascades. Similar receptor-dependent mechanisms for differential activation have been reported for T cells[36]. Therefore, the different strength and persistence of the self-recognition of CLL BcR IGs may influence CLL clinical outcome by modulating the intracellular signal in the pathogenic clones, in turn contributing (at least in part) to the clinical heterogeneity of the disease. To strengthen this point, we analysed using AUC and titration calorimetry a panel of 14 Fab fragments of BcRs from CLL cells that included stereotyped and non-stereotyped cases with indolent and progressive forms of the disease. All Fab fragments from indolent cases show a high-affinity (low $K_D$) self-association, while the ones from patients with aggressive disease display $K_D$ values at least 25-fold greater (Fig. 7). Thus, the quality of the BcR-mediated autologous signal may be directly responsible for the biological features of CLL cells. We anticipate, however, that the CLL BcR intracellular signalling, shaped by the avidity of the homotypic BcR interactions, could be modified by other cellular factors including CD38 expression, a known negative prognostic marker[37], or cell-intrinsic aberrations affecting critical pathways and processes (for example, the p53 pathway). A quantitative analysis of the $Ca^{2+}$ influx in TKO cells expressing the subset no. 2 or subset no. 4 BcR IGs shows a trend for a larger amount of ions mobilized by the former, albeit not reaching statistical significance (Supplementary Fig. 9), which might also be explained by the different activation thresholds and intracellular

signalling motifs of IgM and IgG BcRs[38,39]. Moreover, BcR-transfected TKO cells are a reconstituted system not fully reflecting the situation in primary CLL cells, hence definitive conclusions on the actual *in vivo* effect of the long-term autonomous stimulation in CLL cells cannot be directly extrapolated from these *in vitro* experiments. Future studies should clarify the interplay between direct BcR autonomous signalling and other pathways in shaping the fate of the leukaemic cells.

The finding that BcR IGs are able to initiate intracellular $Ca^{2+}$ influx through self-recognition raises a question on the possible physiological role of similar events in normal B-cell development and maturation. Although the BcRs used in this study are derived from neoplastic B cells, in principle similar homotypic interactions may play a role in providing tonic signals to normal B cells in the mature repertoire. Indeed, the development of CLL likely requires other concurrent factors, such as genetic aberrations, and the autonomous signalling alone may be insufficient to drive the expansion of the leukaemic clones. Further molecular and structural studies will shed further light on the cellular and immunological relevance of the BcR self-recognition.

In summary, the distinctive cell-autonomous signalling for CLL-derived BcRs is a consequence of homotypic interactions established between receptor molecules. Diverse CLL BcR IGs display unique self-epitopes, binding kinetics and affinity, and subsequent functional responses, and this diversity within the homotypic BcR interaction likely underlies the heterogeneity characterizing CLL at the biological and clinical level. Taken together, the present findings offer a novel perspective to BcR antagonism[40] as an attractive therapeutic strategy for CLL, targeting the BcR structure rather than the signalling initiated by BcR triggering. Homology models of CLL BcR IGs previously suggested that receptors from different subsets, as well as

non-stereotyped cases, might be clustered according to the combining site similarity[41]. It will be of extreme interest to see if this insight can be confirmed with experimental structures, thus allowing to design bioactive antagonists that can simultaneously target different subsets of patients carrying stereotyped receptors rather than individual CLL cases.

## Methods

**Protein preparation.** Immunoglobulin heavy, κ, and λ light-chain gene rearrangements of CLL subset no. 4 cases CLL183 and CLL240 (ref. 42) and subset no. 2 case P11475[35] were PCR amplified and fused to the human IgG1 or μ, human κ or human λ constant region, respectively. For the CLL183 case, PCR-based site-directed mutagenesis was performed to generate the epitope (Glu$^{16H}$Ala) and the paratope (Arg$^{107H}$Ala/Tyr$^{108H}$Ala) double mutant (for all primer sequences used, see Supplementary Data 1). Plasmids were maintained in *Escherichia coli* DH5α and TOP10 cells (Invitrogen). All proteins were expressed and purified by SynoGene (San Diego, CA, USA). The Fabs from cases CLL240, CLL183 and the two CLL183 mutants were expressed in *E. coli* BL21(DE3) cells (Invitrogen) via a periplasmic secretion system, while the P11475 IgM Fab fragment was expressed in HEK293 cells (ATCC CRL-1573). Similarly, IgM Fab fragments from subsets no. 1 (P10015), no. 6 (P10824), no. 7 (CLL1430), no. 10 (P5071), no. 77 (P14338, P2959, P5623, P4807), no. 148b (P6340), no. 169 (P6540), and from the non-stereotyped case CLL358 (IGHV1-69) were cloned, and expressed in HEK293 cells. All Fabs were purified through Ni·NTA chromatography, taking advantage of an hexahistidine tag inserted at the C terminus of the heavy chain, and stored in a buffer composed of 50 mM Tris (pH 7), and 100 mM glycine. Size exclusion chromatography was performed on a Superdex 75 10/300 GL column (GE Healthcare Life Sciences) coupled to an AktaPurifier10 FPLC System (GE Healthcare Life Sciences). The column was equilibrated with the elution buffer containing 20 mM Tris (pH 7.4) and 150 mM NaCl, and the samples were eluted with a isocratic elution at 0.5 ml min$^{-1}$.

**Analytical ultracentrifugation.** SV and SE experiments were performed using a Beckman/Coulter XL-I analytical ultracentrifuge with double-sector centrepiece and sapphire windows using absorbance detection. SV experiments were conducted at 72,756*g* at 20 °C. CLL240 and wild-type CLL183 were tested at 0.1, 0.2 and 0.4 mg ml$^{-1}$; CLL183 mutants, and the other Fabs were concentrated up to 1 mg ml$^{-1}$ where the absorbance was < 1.0. All experiments were performed both in storage buffer and PBS. SE experiments on wild-type CLL183 and CLL240 Fabs were performed using protein at 0.1, 0.15 and 0.26 mg ml$^{-1}$ in storage buffer, collecting data at 20 °C at velocities of 5,161, 11,612, 20,644 and 39,030*g*. Equilibrium was considered reached when the difference between subsequent readings was within the instrument noise. The SV and SE data were analysed using the programs SEDFIT and SEDPHAT[43]. For each Fab, the SV data were first analysed to determine the distribution of species present in solution. Next, the SE data were globally fitted (velocities and concentrations) to extract the dissociation constant of the complex. Dissociation rates were obtained from the simultaneous fitting of the SV data collected at three different protein concentrations. AUC runs were carried out in triplicate, and individual experiments are shown in the figures.

**Dilution isothermal titration calorimetry.** Dilution isothermal titration calorimetry (ITC) measurements[44] were performed on a VP-ITC calorimeter at 20 °C. Before performing the experiments, the proteins were extensively dialysed against buffer solution (20 mM Tris (pH 7.4), 100 mM glycine, 100 mM NaCl). Each titration involved 25 injections of each protein, concentrated to the maximum achievable (300–400 μM), into the sample cell containing buffer solution at intervals of 340 s. The titration cell was stirred continuously at 307 r.p.m. The integration of thermograms and buffer titration subtraction were performed with NITPIC[45,46], and the resulting isotherms were analysed with SEDPHAT to determine Fab–Fab binding affinity. Experiments were performed in duplicate.

**Crystallographic analysis.** Crystallization was performed with the hanging drop vapour diffusion method. Crystals of CLL240 were obtained by mixing equal volumes of protein at 5 mg ml$^{-1}$ and precipitant solution (0.2 M Bis-Tris (pH 5.7) and 8% PEG 20K). A single crystal was collected and cryoprotected into precipitant solution enriched with 20% glycerol, flash cooled in liquid nitrogen and maintained at cryogenic temperatures throughout data collection. X-ray diffraction data were collected at beamline ID29 of the European Synchrotron Radiation Facility (ESRF, Grenoble, France) on a Pilatus 6M_F pixel array detector using the oscillation method at a wavelength of 0.9762 Å. Data were indexed, integrated, scaled and reduced to unique reflections using XDS[47]. The measured intensities were converted to structure factor amplitudes using TRUNCATE[48], flagging 5% of the unique reflection for cross-validation that were excluded from all stages of refinement. Initial phases for the CLL240 Fab were obtained with the molecular replacement method as implemented in MOLREP[49] using the *VH4-34*-expressing cold agglutinin IgM (PDB ID code 1DN0) as a search model after truncation of the side chains to the Cβ atom. A CLL240 Fab model was built using ARP/wARP[50],

followed by manual adjustment into 2mFo-DFc and mFo-DFc σ$_A$-weighted electron density maps. The model underwent cycles of restrained energy minimization with a maximum-likelihood target function using PHENIX[51] followed by manual rebuilding into σ$_A$-weighted electron density maps in COOT[52]. Torsion angle non-crystallographic symmetry restraints were applied throughout refinement. Anisotropic motion was modelled by refinement of the TLS parameters[53] for groups of residues identified using the TLSMD server[54]. Solvent molecules were added in spherical residual density peaks > 3σ. CLL183 (4 mg ml$^{-1}$) was crystallized using a precipitant solution composed of 0.1 M sodium citrate (pH 3.5), 0.1 M (NH$_4$)$_2$SO$_4$, 1% hexanediol and 14% PEG 3350. Single crystals were collected and cryoprotected in the precipitant solution after increasing the percentage of PEG 3350 to 34%. The diffraction data for CLL183 were collected on the same beamline with identical setup, integrated using MOSFLM[55] and scaled with AIMLESS[56]. Initial phases for the CLL183 data set were obtained with the molecular replacement method using the refined CLL240 Fab as a search model. The structure was refined following the same procedure described for CLL240, with the initial additional inclusion of external restraints for the main chain atoms based on the refined CLL240 Fab structure that were removed in the later stages of refinement. The P11475 Fab (6 mg ml$^{-1}$) was crystallized from a buffer containing 0.1 M Bis-Tris (pH 5.5), 200 mM MgCl$_2$ and 20% PEG 3350, and cryoprotected by raising the PEG concentration to 34%. Diffraction data were collected at beamline ID23-1 of ESRF with λ = 0.97241 Å, and integrated and reduced to unique intensities using XDS[47] and AIMLESS[56]. The crystal displayed an apparent 422 Laue symmetry, but belonged to the *P4$_1$* tetragonal space group, merohedrally twinned through the (*h*, − *k*, − *l*) operator and a refined twin fraction of 0.28. Initial phases were obtained with the molecular replacement technique using the isolated Fv and Fc fragments from the human rheumatoid factor (PDB ID code 1ADQ) as search models. The structure was refined with the procedure described for the other Fabs using Coot and PHENIX and including individual B-factor and TLS refinement, using a least-squares target function accounting for the merohedral twinning.

For all crystal structures, the limit of the diffraction data to be used in the refinement was decided using the paired refinement technique[57,58]. The stereochemical quality of the models was continuously analysed with MOLPROBITY[59] to identify regions requiring manual intervention. Interaction surfaces and intermolecular contacts were analysed using PISA (http://www.ebi.ac.uk/pdbe/pisa). Figures were generated using the program PyMol (http://pymol.sourceforge.net).

**Cell culture.** TKO cells[26] and Phoenix cells (ATCC CRL-3214) were cultured in Iscove's medium (Biochrom) containing 10 mM L-glutamine (Gibco), 10% heat-inactivated fetal bovine serum (PAN-Biotech) and 100 U ml$^{-1}$ penicillin/streptomycin (Gibco). For culture of TKO cells, the medium was supplemented with 50 μM β-mercaptoethanol (Gibco) and supernatant of J558L mouse plasmacytoma cells stably transfected with a vector encoding murine IL-7 (a kind gift from T Rolink to H Jumaa).

**Plamids and retroviral transduction.** ER$^{T2}$-SLP65 was expressed in TKO cells using a retroviral vector containing tdTomato as expression control. Ig heavy and light chains were expressed using the BiFC vector system. Human VH domains were fused to chimeric constant regions consisting of human γ-1 CH1–CH3 and murine transmembrane–cytoplasmatic tail domains. Retroviral transduction was performed as described[26]. Briefly, Phoenix cells were transfected using GeneJuice Transfection Reagent (Merck Millipore) as recommended by the manufacturer's protocol. Supernatants were collected 48 h post-transfection and used for transduction of TKO cells.

**Ca$^{2+}$ flux analysis.** Measurements of intracellular calcium mobilization were performed as described previously[26]. Per sample, $1 \times 10^6$ cells were loaded with the calcium-sensitive dye Indo-1 AM (Molecular Probes). To characterize the BCR signalling capacity, ER$^{T2}$-SLP65 function was induced by adding 2 μM 4-hydroxy tamoxifen (OHT; Sigma Aldrich) after 40 s of baseline recording. For external stimulation of BCR signalling, 10 μg ml$^{-1}$ anti-human κ (Southern Biotech) was used in addition to OHT. Experiments shown in all figures are representatives of triplicates.

**Data availability.** The coordinates and structure factors of the CLL Fabs have been deposited with the Protein Data Bank, accession codes 5DRX (CLL240), 5DRW (CLL183) and 5IFH (P11475). PDB codes 1ADQ, 1A14, 1A2Y, 1ACT, 1BJ1, 1BZQ, 1DN0, 1E6J, 1EGJ, 1FE8, 1H0D and 1HZH were used in this study. All other data are available from the corresponding authors on reasonable request.

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

## Acknowledgements

Diffraction data were collected at beamlines ID29 and ID23-1 of the European Synchrotron Radiation Facility (ESRF, Grenoble, France), and the high-quality support by all beamline staff is gratefully acknowledged. C.M. and M.D. thank Frédéric M.D. Vellieux for suggestions on data collection strategy. This work was supported by research grants from Fondazione Intesa San Paolo Onlus (to M.D.), Associazione Italiana per la Ricerca sul Cancro AIRC Investigator Grant Nos 17032 (to M.D.), 15189 (to P.G.) and Special Program Molecular Clinical Oncology AIRC 5 per mille no. 9965 (to P.G.); 'MEDGENET' (no. 692298) by the EU (to K.S.); and NIH National Cancer Institute RO1

CA081554 (to N.C.). H.J. acknowledges support by the Deutsche Krebshilfe and Deutsche Forschungsgemeinschaft (TRR130, excellence cluster BIOSS, SFB1074) and ERC advanced grant 694992. CM acknowledges partial support by a fellowship from Italian Foundation for Multiple Sclerosis (FISM fellowship 2012/B/9).

## Author contributions

K.S., P.G. and M.D. initiated the project. N.C. provided the cDNA for the subset no. 4 monoclonal antibodies. M.G. cloned the BcRs, performed preliminary expression of the recombinant Fabs and engineered the site-specific mutants. C.M. designed and performed the SEC, AUC and ITC experiments, the AUC data analysis and crystallized the proteins. C.M. and M.D. measured diffraction data, solved, refined and analysed the crystal structures. M.G., An.A., S.N. and K.S. performed comparative sequence analyses. M.D.-v.M. and Al.A. generated expression constructs and D.S., A.T. and R.Ü. performed the Ca$^{2+}$ influx measurements in TKO cells. H.J., K.S. P.G. and M.D. designed experiments, and supervised research. M.D., K.S. and P.G. wrote the manuscript with contributions from all authors. M.D. and P.G. contributed equally as senior authors of this work.

## Additional information

**Competing interests:** The authors declare no competing financial interests.

