## [Peer review file · Nature Communications]

Reviewers' comments:

Reviewer #1 (Remarks to the Author):

The manuscript entitled "Distinct homotypic B-cell receptor interactions shape the outcome of chronic lymphocytic leukemia" by C. Minici et al. describes the molecular mechanisms involved in the activation of the B cell receptor (BCR) of Chronic lymphocytic leukemia (CLL) cells.

This study shows that two types of BCR characterize CLL subsets #4 and #2. Both are characterized by homotypic interactions which continuously stimulate B cells, nevertheless they result in clinically different forms of disease.

This study shows that CLL subset #4 is associated with IgG molecules that have high affinity interactions with long-lived dissociation rate constants. This type of homotypic interactions lead to stable binding, to functionally anergic cells and to clinically indolent CLL as final outcome. The structural studies reveal the relevance of somatic mutations and IgG isotype in this recognition.

The authors also investigated CLL subset #2, characterized by the presence of IgM antibodies making quite different homotypic binding. These are characterized by much weaker interactions with very fast dissociation, leading to productive activation of B cells and correlating with a clinically aggressive type of CLL.

This latter interaction is associated with precise somatic mutations and very often involves the interaction with an epitope in which the Arg generated by the splice site between the IGLJ3 and IGLC genes is crucial.

The results are presented in a crystalline manner, the figures are informative and allow the immediate understanding of the complexity of the findings. The discussion is appropriate as well as the quoted references.

This study offers an important advancement in understanding the types of homotypic interactions that lead to B cell activation and clinically different types of CLL.

Reviewer #2 (Remarks to the Author):

In this manuscript the authors explore the structural basis of B cell-autonomous B cell receptor (BCR) signaling in chronic lymphocytic leukemia (CLL). Such studies are important as they can, and have, led to the identification of targets for therapeutics for many B cell malignancies. The authors provide convincing evidence that homotypic BCR interactions are different for the BCRs of CLL subset #4 that show an indolent clinical course and subset #2 that show an aggressive clinical course. The #4 BCRs interact through high affinity long-lived interactions between VH CDR3 and VH1 CH1 whereas #2 BCRs interact through low affinity, short lived, interactions between VL CDR1/VL CDR2 and VL FR1 and the VL-CL linker. Additional interesting information is provided concerning the contribution of IgH isotype and somatic hyper mutations to the BCR-BCR interactions. The link to the functional outcome of these structurally defined interactions is the demonstration that both #4 and #2 BCRs expressed in BCR-deficient cells result in

Ca²⁺ fluxes. However, CLL #4 and #2 BCRs do appear to be distinguishable either qualitatively or quantitatively by the Ca²⁺ assay even though the hypothesis is that the quality of the BCR-BCR interaction dictates the behavior of the CLL. Although the data provided is convincing that the BCRs of two different CLL classes interact differently the manuscript would benefit from information concerning the behavior of the BCRs in terms of clustering and assembly of signaling complexes. Such analyses would complete this interesting story.

Reviewer #3 (Remarks to the Author):

This potentially interesting manuscript suggests a structural explanation for how homotypic BCR interactions may dictate the clinical course of CLL. Overall the structures and associated biophysical/functional data is undertaken well, and an appealing narrative is provided where the authors link structural observations in a crystal lattice with potential physiological and clinical relevance. However additional experiments are necessary to fully substantiate the conclusions.

One main concern is that the trend of high affinity=indolent phenotype and low affinity=aggressive phenotype is based entirely on two data points. They have not convincingly shown the generality of this observation. Namely, the authors need to more robustly correlate CLL phenotype with BCR homotypic affinity. For example, demonstrate that multiple BCRs derived from indolent CLL have high affinity and multiple BCRs from aggressive CLL have weak affinity, or even better, investigate BCRs derived from a spectrum of CLL phenotypes. Moreover, the extent of mutants tested in the "high affinity" subtype seemed rather limited, and should be expanded upon. Control mutations in each subtype should also be included.

The Ca²⁺ flux data also needs more consideration. This data, which is crucial, seems only qualitative at best, and indeed is very hard to compare between the two subsets. Which subset induces flux more readily? How is cell surface expression levels controlled in this over-expression system and how do the expression levels correlate with that normally found on cells? Why is it that low affinity homotypic BCR interactions would trigger a more aggressive disease? Is this not a little counter-intuitive? Do the lower affinity BCRs trigger a more intensive/sustained calcium flux than the higher affinity ones? Whilst the authors speculate on this enigma in the discussion, this concept remains speculation and no data is provided to support this contention.

The authors should acknowledge/ test whether other factors may also be at play - for example, the different subsets may lead to differing levels of expression on the cell surface, which may also impact on autonomous signalling.

It would be of interest to know if these interactions would be permissive within the context of FL BCR? Can the authors model how this would look from existing FL Ig structures? Would such modelling suggest the interactions can occur in cis, or in trans? This concept is barely mentioned in the discussion. It is also difficult to ascertain how similar/different the #2 and #4 subsets bind to the ligand - a comparative figure, would be useful here. Is the binding interface mutually exclusive? As such, differing docking modes may relate to different signalling outcomes.

Other points

1. Can the authors comment on why the mutants for all receptors are expressed better than wt (Fig 3)?
2. "Steric clash" (line 188): this is not apparent from the associated figure. Perhaps

mutation of K214gamma would firm up the conclusions from this section.

3. AUC on P11475 Fab: the "peak" observed for the homo-dimer is extremely small. Is a more prominent peak at 5S observed at higher (eg 5-10mg/ml) concentration?

4. The SC values between the two subsets are not significantly different

5. BSA values should be rounded to the nearest 10 - the values quoted are too accurate.

6. It is unclear how many times each experiment (AUC, signalling) was independently undertaken.

7. The data collection statistics for CCL240 is poor - greater satisfactory rationale for using this resolution, or curtailing the resolution is required

Response to referees

Reviewer 1

The results are presented in a crystalline manner, the figures are informative and allow the immediate understanding of the complexity of the findings. The discussion is appropriate as well as the quoted references.

This study offers an important advancement in understanding the types of homotypic interactions that lead to B cell activation and clinically different types of CLL.

Reply. We thank Reviewer 1 for the comments both on the quality of the work and on the clarity of the manuscript.

Reviewer 2

In this manuscript the authors explore the structural basis of B cell-autonomous B cell receptor (BCR) signaling in chronic lymphocytic leukemia (CLL)..... However, CLL #4 and #2 BCRs do appear to be distinguishable either qualitatively or quantitatively by the Ca²⁺ assay even though the hypothesis is that the quality of the BCR-BCR interaction dictates the behavior of the CLL.

Reply. We thank Reviewer 2 for appreciating the novelty and relevance of the structural, biochemical and functional analysis of the CLL-derived BcRs.

Regarding the concern on the lack of quantitative comparison of the Ca²⁺ influx in the TKO cells transfected with the subset 2 and 4 BcRs, we do agree that this point required further analysis and we now quantified the differences in the Ca²⁺ influx in the TKO cells. As can be appreciated in the attached figures, for both subset 2 and subset 4 BcRs, the mutants affecting autonomous signaling induce a significantly different Ca²⁺ mobilization (Supplementary Figure 8).

We also performed experiments to compare the Ca²⁺ mobilization between subset 2 and subset 4 BcR-transfected TKO cells. Although a trend for larger Ca²⁺ influx in subset 2-expressing cells is apparent, statistical significance was not reached. This finding likely points out that the early events in BcR-mediated signaling are not affected by the kinetics of the interaction in this model system. In addition, we have to consider that BcR-transfected TKO cells are a reconstituted system not fully reflecting the situation in primary CLL cells, hence definitive conclusions on the actual *in vivo* effect of the long-term autonomous stimulation in CLL cells cannot be drawn. We conclude that future studies should employ primary CLL cells in order to fully appreciate the interplay between BcR signaling and additional factors for defining the clinical course of the disease. A supplementary figure (Supplementary Fig. 8) has been added, together with a discussion of the result (page 13).

Although the data provided is convincing that the BCRs of two different CLL classes interact differently the manuscript would benefit from information concerning the behavior of the BCRs in terms of clustering and assembly of signaling complexes. Such analyses would complete this interesting story.

Reply. We agree with the Reviewer that an analysis of the architecture of the BcR signaling complex at the cell surface would be of great potential interest. Such project, however, would require a dedicated experimental effort to obtain a characterization that we feel is outside the scope of the present manuscript. Indeed, the main, novel conclusions of our work from the structural and the biochemical analysis reported in the revised manuscript are the first

observation of BcR homotypic interactions, and the strong link between their specific features and the intracellular signaling in CLL B cells, and thus to features of the disease.

Reviewer 3

This potentially interesting manuscript suggests a structural explanation for how homotypic BCR interactions may dictate the clinical course of CLL. Overall the structures and associated biophysical/functional data is undertaken well, and an appealing narrative is provided where the authors link structural observations in a crystal lattice with potential physiological and clinical relevance. However additional experiments are necessary to fully substantiate the conclusions.

Reply. We thank Reviewer 3 for the positive reception of the novelty and relevance of our study, and for the in-depth analysis of the manuscript. Prompted by his/her comments, we clarified the points that we failed to convey in optimal fashion in the original submission.

One main concern is that the trend of high affinity=indolent phenotype and low affinity=aggressive phenotype is based entirely on two data points. They have not convincingly shown the generality of this observation. Namely, the authors need to more robustly correlate CLL phenotype with BCR homotypic affinity. For example, demonstrate that multiple BCRs derived from indolent CLL have high affinity and multiple BCRs from aggressive CLL have weak affinity, or even better, investigate BCRs derived from a spectrum of CLL phenotypes.

Reply. We do agree that in order to make a generalized statement on the trend of high affinity=indolent phenotype and low affinity=aggressive phenotype we would need multiple additional cases. However, we feel that this extensive characterization goes beyond the scope of the current work where we aimed at offering proof of principle that the autonomous signaling is characterized by homotypic BcR interactions that may have different structural features, affinities and half-lives in different CLL cases. Our findings provide an explanation for the known heterogeneity of the disease and that all the aforementioned factors are crucial determinants of the B cell clonal response, and eventual clinical behavior.

Our study unleashes the concept that autonomous signaling is not a “One-size-fits-all” phenomenon but instead may be differentially shaped in different cases. In order to reveal this and circumvent the incapacitating heterogeneity of CLL, we zoomed-in on paradigmatic subgroups with homogeneous biological features and clinical outcome represented by the two stereotyped subsets described in the paper: these subsets stand at the extreme opposite ends of the disease spectrum, getting a rather “black/white” readout. However, and in line with the reasoning of the reviewer, we neither believe nor claim that the trend in terms of responses will be simply dichotomic when assessing more cases even chosen among the aggressive or the indolent ones but outside the two studied subsets. Studying more cases will likely mirror the great heterogeneity of CLL where other contributing factors (e.g. genetic lesions, DNA methylation changes, and altered microRNA profiles) are also asymmetrically distributed between CLL cases with distinct clinical outcome, thus potentially enhancing or dampening the affinity/outcome effect. Analyzing this variability in detail and assessing whether our observations may serve as a basis for a novel prognostic/predictive functional assay is ongoing in our group but will need tens if not hundreds of immunoglobulins from well characterized patients and will be palatable for a more clinical audience in the distant future.

Moreover, the extent of mutants tested in the "high affinity" subtype seemed rather limited, and should be expanded upon. Control mutations in each subtype should also be included.

Reply. We thank the reviewer for this comment that refers to the mutations analyzed in subset 4. Besides the 2 previous mutations affecting the paratope (HCDR3 loop) and the epitope (FR1 region) and unambiguously confirming that the mode of interaction observed in the crystal structure reflects the contacts taking place in solution, we now included a control mutation of an amino acid in the HCDR3 that is not involved in the intermolecular contacts and indeed does not affect Ca^{2+} influx. In addition, also in light of the minor comment of this reviewer on the role of the $\text{C}\gamma 1$ domain, we also constructed a chimeric subset 4 BcR, where the constant domains of CLL183 were replaced by the murine $\text{C}\gamma 1/\text{C}\kappa$ pair (page 8). The mouse domain lacks the Lys214 residue, while retaining a similar conformation of the loop region. This receptor was expressed in TKO cells, and was not capable of autonomous signaling, thus completing the mutagenesis study of the subset 4. Indirectly, this result shows that the integrity of the IgG domain is required for subset 4 autonomous signaling, suggesting that the isotype switch to IgG is a key event in the development of the disease in this particular group of patients.

The Ca^{2+} flux data also needs more consideration. This data, which is crucial, seems only qualitative at best, and indeed is very hard to compare between the two subsets. Which subset induces flux more readily?

Reply. Following the questions posed by the reviewers, we analyzed in a quantitative manner the Ca^{2+} influx. We now show that the BcR mutants engineered are less effective in inducing Ca^{2+} influx. Moreover, there is a trend for BcRs derived from the aggressive subset 2 cases to mobilize more Ca^{2+} compared to the ones derived from the indolent subset 4 cases, albeit the difference did not reach statistical significance. We interpret this result as a potential consequence of at least two different factors, 1) Ca^{2+} influx is an early event in BcR-mediated signaling consequence of the BcR declustering, thus independent of the binding affinity; 2) BcR-transfected TKO cells are a reconstituted system not fully reflecting the situation in primary CLL cells, hence definitive conclusions on the effect of the long term autonomous stimulation in CLL cells cannot be drawn. A relevant comment has been added to the revised Discussion to acknowledge this fact, as suggested by the Reviewer (page 13).

How is cell surface expression levels controlled in this over-expression system and how do the expression levels correlate with that normally found on cells? Why is it that low affinity homotypic BCR interactions would trigger a more aggressive disease? Is this not a little counter-intuitive? Do the lower affinity BCRs trigger a more intensive/sustained calcium flux than the higher affinity ones? Whilst the authors speculate on this enigma in the discussion, this concept remains speculation and no data is provided to support this contention. The authors should acknowledge/test whether other factors may also be at play - for example, the different subsets may lead to differing levels of expression on the cell surface, which may also impact on autonomous signaling.

Reply. We apologize for not being more explicit on this delicate issue, and we now clarified in the revised discussion the concept that BcR binding to high affinity antigens leads to anergy (page 12, see also references 31 (Diz et al., J Immunol 2008) and 32 (Taylor et al., J Exp Med 2012). Moreover, we draw a parallel with T cells, whose sustained signaling is mediated by serial triggering events where a single antigen (peptide/MHC) engages sequentially with different T cell receptors to provide a sustained signal, as shown by the group of Lanzavecchia (page 13, reference 34). Similarly, antigens that induce short lived complexes at the cell surface can deliver a time-pulsed signal that leads to a continuous response by the B cell. Conversely, a long lived

complex delivers a qualitatively different signal that activates anergy mechanisms. This is now stated in the revised discussion, page 13. Regarding the levels of BcR expression, that are indeed different in different subsets (anergic vs responding CLL) of the disease, one has again to remember that BcR-transfected TKO cells are a reconstituted system and the levels of the surface molecules indeed cannot be controlled apart from flow cytometry, thus perhaps limiting, in this respect, an actual representation of what is happening in vivo. This lack of difference in terms of levels of expression may also contribute to better understanding the inconclusive results on the quantitation of Ca²⁺ flux as described above (Supplementary Figure 8).

It would be of interest to know if these interactions would be permissive within the context of FL BCR? Can the authors model how this would look from existing FL Ig structures? Would such modelling suggest the interactions can occur in cis, or in trans? This concept is barely mentioned in the discussion. It is also difficult to ascertain how similar/different the #2 and #4 subsets bind to the ligand - a comparative figure, would be useful here. Is the binding interface mutually exclusive? As such, differing docking modes may relate to different signalling outcomes.

Reply. Following the reviewer's suggestions on a full length (FL) BcR, we have now expanded the discussion, stating that:

- a. The mode of interaction is compatible with both inter-cell (*trans*) and intra-cell (*cis*) recognition (page 12). We provide a supplementary figure (Supplementary Figure 6, panels a, b, and c) with the model for subset 4, using as a template one of the FL IgG structures available in the PDB. It should be noted that the two experimental structures available differ dramatically in the orientation of the Fab fragments with respect to the Fc portion, demonstrating a high flexibility of the molecule that can further facilitate the homotypic interaction both in *cis* and *trans*. The same modelling exercise was carried out for subset 2, and it is presented with the caveat that no FL IgM structures have yet been determined. Yet, since the overall features of the BcR Ig structure (two Fab fragments linked via flexible regions to the CH portions) is conserved, the same accessibility of the epitope is predicted in IgMs. The model presented supports *cis* interactions. However, given the lack of FL IgM structures and the flexibility of the Fab fragments, it is difficult with the present knowledge to assess whether the differences between the two FL BcR models reflect actual differences at the cell surface which may be responsible, as the reviewer punctually stated, for the differences in intracellular signaling.
- b. We provide a Supplementary Figure (Supplementary Figure 7, panel a and b) with a side-by-side comparison of the homotypic interactions of the subset 2 and subset 4 BcRs. The "antigen" molecule is oriented vertically in both panels. The figure provides a clear, visual confirmation that the interaction surfaces are distinct (for subset 2, the linker region of the lambda chain; for subset 4, a composite region spanning FR1 and Cγ1).

Other minor points:

1. *Can the authors comment on why the mutants for all receptors are expressed better than wt (Fig 3)?*

Reply. The higher expression of the BcR mutants can be explained by the fact that the signaling-competent complexes are more readily internalized in the B cells.

2. *"Steric clash" (line 188): this is not apparent from the associated figure. Perhaps mutation of K214gamma would firm up the conclusions from this section.*

Reply. In order to address the reviewer's concern, we now provide an experimental proof of the structure-based induction of the role of the C γ 1 constant domain in forming the epitope, by constructing a chimeric subset 4 BcR, where the constant domains of CLL183 were replaced by the murine C γ 1/C κ pair. The murine C γ 1 domain contains a Pro residue in place of Lys214, while retaining a highly similar conformation (0.65 Å root mean square distance for 77 C α atoms). This receptor was expressed in TKO cells and is not capable of autonomous signaling (new Figure 3c). Thus, the C γ 1 domain is necessary for the formation of the conformational epitope involved in homotypic BcR interaction in the subset 4 CLL.

3. AUC on P11475 Fab: the "peak" observed for the homo-dimer is extremely small. Is a more prominent peak at 5S observed at higher (eg 5-10mg/ml) concentration?

Reply. Unfortunately, concentrations of P11475 exceeding 1.0 mg/mL result in absorbance values > 1.0, enhancing the photometric error and yielding sedimentation velocity profiles that cannot be appropriately fitted. Moreover, the variation on viscosity of the protein solution also hinders the experiment, namely the comparison of the values obtained at different concentrations. The SV experiment reported has been repeated three times, and the appearance of the peak at 5 S is reproducible and consistent, thus we are confident that it does indeed represent a dimeric species, albeit characterized by a large dissociation constant.

4. The SC values between the two subsets are not significantly different

Reply. We agree with the reviewer that these values are not strikingly different, also taking into account that differences in model accuracy may influence the Sc value. In the revised text, we now removed any hint that we overestimate the individual importance of the Sc values reported (page 9) as, indeed, we do not wish to link the low affinity of the P11475 BcR to this value. The message from the structural analysis is that the higher affinity of the subset 4 Fab fragments is linked to the combined effect of number of contacts, buried surface area, and surface shape complementarity. We feel that showing the trend of all surface parameters gives the reader a comprehensive view of the interface properties.

5. BSA values should be rounded to the nearest 10 - the values quoted are too accurate.

Reply. We agree with the reviewer that the buried surface area is at best measurable with a 10 Å² accuracy, and the reason why we used more digits is to avoid confusion for the reader who wishes to reproduce the values. Indeed, these values were taken directly from the output of PISA. Since the BSA values in the manuscript are used as a comparison of the BcR-BcR interactions with other Fab-antigen complexes, we would rather keep the values reported but we are willing to follow the reviewer's indications if there is a strong feeling about it. Our choice is not unusual, see for instance Extended Data Table 2 in Rouvinski et al. (2015) Nature 520, 109-113.

6. It is unclear how many times each experiment (AUC, signalling) was independently undertaken.

Reply. We apologize for not being precise in reporting this data as indeed the numbers were missing. We now added a statement in the methods section that the data shown are representative of experiments performed in triplicate (pages 15 and 19).

7. The data collection statistics for CCL240 is poor - greater satisfactory rationale for using this resolution, or curtailing the resolution is required.

Reply. The reviewer posed a legitimate question and these details should be explicit in the text. Thus, we added a short sentence stating that the paired refinement technique (referenced) was used to determine the high resolution cutoff for inclusion of the data in refinement (page 17). In summary, in order to decide the cutoff for the reflections to be included in the refinement we followed the procedures outlined by Karplus and Diederichs (Karplus PA and Diederichs K (2012) Science 336, 1030-1033, and Diederichs K and Karplus PA (2013) Acta Cryst sect D 69, 1215-1222). First, we used a maximum resolution limit for the scaling and merging of the integrated data using a 0.3 cutoff for the $CC_{1/2}$ correlation coefficient in the highest resolution shell. Then, we carried out a paired refinement procedure to determine whether the weak high resolution data did contribute to a better structural model. For CLL240, we used the model refined against all data to 2.1 Å to calculate R and Rfree using datasets truncated to 2.2, 2.3, 2.4, and 2.5 Å (without refinement). Then, we refined the model against the truncated datasets using phenix.refine with the same protocol (3 macrocycles including positional, TLS, and B-factor refinement). In all instances, the model refined to 2.1 Å has the smallest Rfree-R gap, thus showing less overrefinement. Moreover, the Rfree value calculated from the model refined to 2.1 Å at a given resolution is either smaller or less than 0.05% different from the one derived from the refinement. For clarity, we include (for review purpose only) a table showing the results of the paired refinement for CLL240.

Resolution	R	Rfree	Rfree-R	R	Rfree	Rfree-R	ΔR	ΔR_{free}
	Calculated from 2.1 Å model			Refined			R-C	
2.1	-	-	-	0.1973	0.2206	0.0233	-	-
2.2	0.1938	0.2170	0.0232	0.1918	0.2201	0.0283	-0.002	0.0031
2.3	0.1911	0.2131	0.022	0.1742	0.2126	0.0384	-0.0169	-0.0005
2.4	0.1888	0.2107	0.0219	0.1777	0.2101	0.0324	-0.0111	-0.0006
2.5	0.1871	0.2076	0.0205	0.178	0.2087	0.0307	-0.0091	0.0011

Reviewers' comments:**Reviewer #1 (Remarks to the Author):**

The manuscript has been extensively improved. The new discussion and the addition of new figures confirm the data, make the overall flow of the text clearer than the previous version and provide adequate information on the minor points missing in the originally submitted manuscript.

Reviewer #2 (Remarks to the Author):

The authors have adequately addressed my comments.

Reviewer #3 (Remarks to the Author):

In the revision, the authors have provided very limited additional data to substantiate the main conclusions of the paper. The major critique of $n=2$ data points was not addressed experimentally, with the preferred approach of arguing the point. A similar situation arises with many of the other points raised, and where new experimental data was provided, it was inconclusive (eg Ca^{2+} flux). Given the centrality of the critiques to the main message, the authors revision is left wanting.

Response to the reviewers

Reviewer 1 and Reviewer 2

We thank the reviewers for their appreciation of the improvements of the manuscript and the completeness of the reported work. In this revised manuscript, we present a widening of the biochemical study by analysing a wider panel of BcR Fab fragments, confirming stronger affinity between receptors from indolent cases compared to the aggressive ones (depicted graphically in Figure 7, and discussed on page 13 of the revised manuscript).

Reviewer 3

Below we provide a point-by-point response of the original and second-round reviews, in particular underscoring all experimental actions that we have performed to address all the issues raised.

First round of comments: "One main concern is that the trend of high affinity=indolent phenotype and low affinity=aggressive phenotype is based entirely on two data points. They have not convincingly shown the generality of this observation. Namely, the authors need to more robustly correlate CLL phenotype with BCR homotypic affinity. For example, demonstrate that multiple BCRs derived from indolent CLL have high affinity and multiple BCRs from aggressive CLL have weak affinity, or even better, investigate BCRs derived from a spectrum of CLL phenotypes."

Second round of comments: "In the revision, the authors have provided very limited additional data to substantiate the main conclusions of the paper. The major critique of n=2 data points was not addressed experimentally, with the preferred approach of arguing the point."

Reply. We apologize with the reviewer for not fully addressing this major concern in the first round, and we are now providing novel experimental data on the affinity of a total of 14 recombinant BcR Fab fragments produced, as part of our ongoing work, from 8 aggressive and 6 indolent CLL patients. We used both analytical ultracentrifugation and dilution isothermal titration calorimetry on all 14 recombinant BcR Fab fragments to measure the affinity of the homotypic interactions. We show (page 13 and the new Figure 7) that the BcR Fabs from indolent cases have low K_D values, between 0.7 and 16 μM , while the ones from aggressive cases display much weaker homotypic affinity (>400 μM). These novel results further support the proposed association between homotypic interaction avidity and the clinical course of the disease likely through influencing the anergic vs. responsive phenotype of the CLL cells.

Moreover, the extent of mutants tested in the "high affinity" subtype seemed rather limited, and should be expanded upon. Control mutations in each subtype should also be included.

Reply. Following the reviewer's request, the number of mutants in the subset 4 cases has been expanded. In detail, we mutated the constant domain by swapping with the murine counterpart to demonstrate the role of Lys^{214H} in the epitope (page 8). Together with the Arg^{107H}Ala/Tyr^{108H}Ala double mutant and the Glu^{16H}Ala mutant we mutated all residues that contribute at a significant level to the binding energy according to an analysis with the program FoldX. Thus, now we analysed a total of four mutants confirming that the interactions observed in the crystal structure are necessary for autonomous signalling. As also requested by the reviewer, a control mutation (Thr^{102H}Ala) was also included (page 7). The results confirmed the previous findings, and are included in the revised version of the manuscript (Supplementary Figure 3).

The Ca²⁺ flux data also needs more consideration. This data, which is crucial, seems only qualitative at best, and indeed is very hard to compare between the two subsets. Which subset induces flux more readily?

Reply. Prompted by the reviewer's suggestion, we performed a quantitative analysis of Ca²⁺ influx (reported as Supplementary Figure 8 in the first revision) revealing that, overall, the less stable homotypic interactions between subset #2 BcRs are associated with a trend for larger mobilization of Ca²⁺ compared to the stable intermolecular interactions in subset #4. The inability to reach statistical significance, though contradicting the reviewer's assumption that a difference in affinity between two different BCRs might translate into a difference in Ca²⁺ influx, is not unexpected based on the current literature. Indeed, the studied BcRs possess different constant regions (IgM vs IgG) and hence have different cell surface localization, different activation thresholds, and intracellular signalling motifs (i.e. IgG tail. See Engels et al. (2009) *Nat Immunol* 10, 1018-1025, or Maity et al. (2015) *Sci Signal* 8, 394). Also, the cell system used in our experiments cannot fully recapitulate the setting of a CLL B cell, and the Ca²⁺ influx has to be mainly considered as a highly sensitive readout for the presence of autonomous signalling consequential to the homotypic BcR interactions. For all these reasons, we believe that the lack of statistical significance in the Ca²⁺ mobilization does not negate the biological implications of our reported findings on the differences in affinity and binding between CLL cases with opposite biological and clinical features. A comment acknowledging this together with the relevant references here indicated has been included in the manuscript (page 13).

Why is it that low affinity homotypic BCR interactions would trigger a more aggressive disease? Is this not a little counter-intuitive? Do the lower affinity BCRs trigger a more intensive/sustained calcium flux than the higher affinity ones? Whilst the authors speculate on this enigma in the discussion, this concept remains speculation and no data is provided to support this contention. The authors should acknowledge/ test whether other factors may also be at play - for example, the different subsets may lead to differing levels of expression on the cell surface, which may also impact on autonomous signalling.

Reply. We apologize if we were not absolutely clear on this issue, and we tried to make this concept more explicit in the current version. Indeed, though the reported low affinity homotypic interactions/aggressive disease association might at first glance appear counter-intuitive, it is fully in line with the well-established facts that (i) strong antigen binding and intense signalling induces energy in B cells, (ii) anergy and diminished signalling capacity characterize indolent CLL cases. We now highlighted the references that demonstrated a correlation between B cell anergy and antigen binding avidity (refs. #31 Diz *et al.*, and #32, Taylor *et al.*), and the anergic phenotype of CLL cells from indolent cases (refs. 16 Ntoufa *et al.*, #33 Apollonio *et al.*, and #34 Muzio *et al.*). That notwithstanding, we certainly agree with reviewer 3 that other factors may modulate the clinical history of CLL, and for this reason added in the first revision a relevant comment on page 13.

It would be of interest to know if these interactions would be permissive within the context of FL BCR? Can the authors model how this would look from existing FL Ig structures? Would such modelling suggest the interactions can occur in cis, or in trans? This concept is barely mentioned in the discussion. It is also difficult to ascertain how similar/different the #2 and #4 subsets bind to

the ligand - a comparative figure, would be useful here. Is the binding interface mutually exclusive? As such, differing docking modes may relate to different signalling outcomes.

Reply. As requested by reviewer 3, we produced models of the full length BcRs and showed that they are permissive with intercellular and intracellular interactions. We also included a comparative figure to highlight the different geometries of homotypic interactions in the subset 2 IgM and subset 4 IgG receptors. These analyses are included as Supplementary Figures 6 and 7 in the revised version.

Other minor points:

1. Can the authors comment on why the mutants for all receptors are expressed better than wt (Fig 3)?

Reply. The higher expression of the BcR mutants can be explained by the fact that the signalling-competent complexes are more readily internalized in the B cells.

2. "Steric clash" (line 188): this is not apparent from the associated figure. Perhaps mutation of K214gamma would firm up the conclusions from this section.

Reply. In order to address the reviewer's concern, we constructed a chimeric subset 4 BcR where the constant domains of CLL183 were replaced by the murine C γ 1/C κ pair (reported on page 8) that provided an experimental proof of the structure-based induction of the role of the C γ 1 constant domain in forming the epitope. The murine C γ 1 domain contains a Pro residue in place of Lys214, while retaining a highly similar conformation (0.65 Å root mean square distance for 77 C α atoms). This receptor was expressed in TKO cells and is not capable of autonomous signalling (Figure 4c). Thus, the integrity of the C γ 1 domain is necessary for the formation of the conformational epitope involved in homotypic BcR interaction in the subset 4 CLL.

3. AUC on P11475 Fab: the "peak" observed for the homo-dimer is extremely small. Is a more prominent peak at 5S observed at higher (eg 5-10mg/ml) concentration?

Reply. The suggestion by the reviewer is in principle correct, and indeed we attempted such approach from the early stages of the study. Concentrations of P11475 exceeding 1.0 mg/mL resulted in absorbance values > 1.0, giving unacceptable photometric errors and yielding sedimentation velocity profiles that could not be appropriately fitted. Moreover, the variation on viscosity of the protein solution also hindered the SV-AUC experiment, not allowing the comparison of the values obtained at different concentrations. The SV-AUC experiment reported has been repeated three times, and the appearance of the peak at 5 S is reproducible and consistent, thus we were confident that it did indeed represent a dimeric species, albeit characterized by a large dissociation constant. In this revised version we provide a measurement of the dissociation constant using dilution isothermal titration calorimetry (Figure 5d), that indeed confirms a very weak affinity (430 μ M). We wish to stress that weak protein-protein interaction can pose a formidable challenge in affinity measurements, since the concentrations required for optimal affinity measurements range between 0.1 and 10 \cdot K_D (for P11475, 10 \cdot K_D corresponds to 210 mg/mL). For weakly-interacting systems, it is often impossible due to solubility limits to reach such high concentrations. In the case of dilution ITC, the amount of heat released or absorbed in each injection is also a limiting factor. Despite the inherent limitations in direct affinity

measurements for the weakly-interacting subset 2 BcR Fabs, our mutagenesis studies in the cellular system confirm the nature and specificity of the homotypic interactions.

4. The Sc values between the two subsets are not significantly different

Reply. We agree with the reviewer that these values are not strikingly different, also taking into account that differences in model accuracy may influence the Sc value. In the revised text, we now removed any hint that we overestimate the individual importance of the Sc values reported (page 9) as, indeed, we do not wish to link the low affinity of the P11475 BcR to this value. The message from the structural analysis is that the higher affinity of the subset 4 Fab fragments is linked to the combined effect of number of contacts, buried surface area, and surface shape complementarity. We feel that showing the trend of all surface parameters gives the reader a comprehensive view of the interface properties.

5. BSA values should be rounded to the nearest 10 - the values quoted are too accurate.

Reply. We agree with the reviewer that the buried surface area is at best measurable with a 10 Å² accuracy, and the reason why we used more digits is to avoid confusion for the reader who wishes to reproduce the values. Indeed, these values were taken directly from the output of PISA. Since the BSA values in the manuscript are used as a comparison of the BcR-BcR interactions with other Fab-antigen complexes, we would rather keep the values reported but we are willing to follow the reviewer's indications if there is a strong feeling about it. Our choice is not unusual, see for instance Extended Data Table 2 in Rouvinski et al. (2015) Nature 520, 109-113.

6. It is unclear how many times each experiment (AUC, signalling) was independently undertaken.

Reply. We apologize for not being precise in reporting this data as indeed the numbers were missing. We now added a statement in the methods section that the data shown are representative of experiments performed in triplicate (pages 15 and 19). The dilution ITC measurements were repeated twice (page 16-17).

7. The data collection statistics for CCL240 is poor - greater satisfactory rationale for using this resolution, or curtailing the resolution is required.

Reply. We added a short sentence stating that the paired refinement technique (referenced, page 19) was used to determine the high resolution cutoff for inclusion of the data in refinement (page 17). In summary, in order to decide the cutoff for the reflections to be included in the refinement we followed the procedures outlined by Karplus and Diederichs (Karplus PA and Diederichs K (2012) Science 336, 1030-1033, and Diederichs K and Karplus PA (2013) Acta Cryst sect D 69, 1215-1222). First, we used a maximum resolution limit for the scaling and merging of the integrated data using a 0.3 cutoff for the CC_{1/2} correlation coefficient in the highest resolution shell. Then, we carried out a paired refinement procedure to determine whether the weak high resolution data did contribute to a better structural model. For CLL240, we used the model refined against all data to 2.1 Å to calculate R and R_{free} using datasets truncated to 2.2, 2.3, 2.4, and 2.5 Å (without refinement). Then, we refined the model against the truncated datasets using phenix.refine with the same protocol (3 macrocycles including positional, TLS, and B-factor refinement).

In all instances, the model refined to 2.1 Å has the smallest R_{free}-R gap, thus showing less overrefinement. Moreover, the R_{free} value calculated from the model refined to 2.1 Å at a given resolution is either smaller or less than 0.05% different from the one derived from the refinement.

For clarity, we include (for review purpose only) a table showing the results of the paired refinement for CLL240. Hence, the inclusion of all data measured to 2.1 Å is justified by the attainment of a better refined model.

Resolution	R	Rfree	Rfree-R	R	Rfree	Rfree-R	ΔR	ΔR_{free}
	Calculated from 2.1 Å model			Refined			Ref-Calc	
2.1	-	-	-	0.1973	0.2206	0.0233	-	-
2.2	0.1938	0.2170	0.0232	0.1918	0.2201	0.0283	-0.002	0.0031
2.3	0.1911	0.2131	0.022	0.1742	0.2126	0.0384	-0.0169	-0.0005
2.4	0.1888	0.2107	0.0219	0.1777	0.2101	0.0324	-0.0111	-0.0006
2.5	0.1871	0.2076	0.0205	0.178	0.2087	0.0307	-0.0091	0.0011